

# The impacts of moisture transport on drifting snow
# sublimation in the saltation layer
**N. Huang and X. Dai**
Key Laboratory of Mechanics on Disaster and Environment in Western China,
Lanzhou University, Lanzhou 730000, China
Corresponding to: N. Huang (huangn@lzu.edu.cn)





**Abstract**
Drifting snow sublimation (DSS) is an important physical process related to moisture
and heat transfer that happens in the atmospheric boundary layer, which is of
glaciological and hydrological importance. It is also essential in order to understand
the mass balance of the Antarctic ice sheets and the global climate system. Previous
studies mainly focused on the DSS of suspended snow and ignored that in the
saltation layer. Here, a drifting snow model combined with balance equations for heat
and moisture is established to simulate the physical DSS process in the saltation
layer. The simulated results show that DSS can strongly increase humidity and
cooling effects, which in turn can significantly reduce DSS in the saltation layer.
However, effective moisture transport can dramatically weaken the feedback effects.
Due to moisture advection, DSS rate in the saltation layer can be several orders of
magnitude greater than that of the suspended particles. Thus, DSS in the saltation
layer has an important influence on the distribution and mass-energy balance of snow
cover.



# 1 Introduction

Drifting snow is a special process of mass-energy transport in the hydrological cycle of snow. It not only changes the snow distribution but also results in phase changes of ice crystals into water vapor, which is known as DSS. Snow sublimation not only significantly influences the mass-energy balance of snow cover (e.g., Zhou et al., 2014) by changing surface albedo (Allison, 1993) and the runoff of snowmelt in cold regions (Marks and Winstral, 2001), but also has a pivotal status on moisture and heat transfer in the atmospheric boundary layer(Pomeroy and Essery, 1999; Anderson and Neff, 2008). Thus, it is of glaciological and hydrological importance (Sugiura and Ohata, 2008). In high cold area, the reduction of snow cover may cause the surface temperature to increase in the cold season(Huang et al., 2008, 2012). The thickness of seasonally frozen ground has decreased in response to winter warming (Huang et al., 2012). On the other hand, both dust and biomass burning aerosols may impact the surface albedo when deposited on snow; soot in particular has large impacts on absorption of radiation (Huang et al., 2011). In addition, a large, but unknown, fraction of the snow that falls on Antarctica is removed by the wind and subsequently sublimates. Therefore, a detailed knowledge of DSS is also essential in order to understand snow cover distribution in cold high area as well as the mass balance of the Antarctic ice sheets, and further the global climate system (Yang et al., 2010).

In drifting snow, snow particles can experience continuous sublimation, which induces a heat flux from the surrounding air to the particle and a moisture flux in the opposite direction (Bintanja, 2001a). Thus, DSS can cause increases in humidity and cooling of the air (Schmidt, 1982; Pomeroy et al., 1993) and has an inherent self-limiting nature due to the feedback associated with the heat and moisture budgets (Déry and Yau, 1999; Groot Zwaaftink et al., 2011, 2013). On one hand, snow sublimation absorbs heat and decreases the temperature of the ambient air, which in turn reduces the saturation vapor pressure and hence the sublimation rate; on the other hand, the increment in the moisture content of the ambient air decreases the sublimation rate of drifting snow, as it is proportional to the under-saturation of the air.

Saltation is one of the three modes of particle motion, along with suspension and creep. Among the three modes, saltation is important and the DSS in the saltation





layer may constitute a significant portion of the total snow sublimation (Dai and
Huang, 2014). Previous studies of DSS mostly focused on the sublimation of
suspended snow, which was mainly due to the consideration that sublimation will
soon vanish in the saltation layer because the feedback of DSS may lead to a
saturated layer near the surface (Bintanja, 2001b). However, the field observation
data of Schmidt (1982) showed that relative humidity only slightly increases during
snowdrift events and the maximum humidity was far below saturation. Further
studies (Groot Zwaaftink et al., 2011; Vionnet et al., 2013) also showed that the
relative humidity does not reach saturation even at the lowest atmosphere level after
DSS occurs. Some scientists argued that it was caused by moisture transport, such as
diffusion and advection of moisture, which inevitably accompany the drifting snow
process (Vionnet et al., 2013). Therefore, it is necessary to study the feedback
mechanism of DSS in the saltation layer and the effect of moisture transport on it.
In this study, a wind-blown snow model, balance equations for heat and moisture
of an atmospheric boundary layer, and an equation for the rate of mass loss of a
single ice sphere due to sublimation were combined to study the sublimation rate of
drifting snow by tracking each saltating particle in drifting snow. Then, the effects of
DSS on the humidity and temperature profiles, as well as the effects of diffusion and
advection of moisture on DSS in the saltation layer, were explored in detail.

**2 Methods**
**2.1 Model Description**
Saltation can be divided into four interactive sub-processes, i.e., aerodynamic
entrainment, particle trajectories, particle-bed collisions, and wind modification
(Huang et al., 2011).
The motion equations for snow particles are (Huang et al., 2011)
$$m_p \frac{dU_p}{dt} = F_D \left( \frac{U_f - U_p}{V_r} \right), \qquad (1)$$

$$m_p \frac{dV_p}{dt} = -W_g + F_B + F_D \left( \frac{V_f - V_p}{V_r} \right), \qquad (2)$$





$$\frac{dx_p}{dt} = U_p ,$$
(3)

$$\frac{dy_p}{dt} = V_p .$$
(4)

where $m_p$ and $W_g$ are the mass and weight of the snow particle, respectively; $U_f, V_f$,
$U_p$ and $V_p$ are the horizontal and vertical velocities of the airflow and snow particle,
respectively; $V_r = \sqrt{\left(U_f - U_p\right)^2 + \left(V_f - V_p\right)^2}$ is the relative velocity between the
airflow and snow particle; $x_p$ and $y_p$ are the horizontal position and vertical height of
the snow particle, respectively; $F_B = \frac{1}{6}\rho_f \pi D^3 g$ and $F_D = \frac{1}{8}C_D\rho_f \pi D^2 V_r^2$ are the
buoyancy force and the drag force applied on the snow particle, respectively; $\rho_f$ is the
air density; $D$ is the diameter of the snow particle; $g$ is the acceleration of gravity; and
$C_D$ is the drag coefficient.

11        Within the atmospheric boundary layer, the mean horizontal wind

velocity $u$ satisfies the Navier-Stokes equation (Werner, 1990). According to
Prandtl's mixing length theory for the steady flow fully developed over an infinite
planar bed, $u$ is

$$\frac{\partial}{\partial y}(\rho_f \kappa^2 y^2 \left|\frac{du}{dy}\right|\frac{du}{dy}) + F_x = 0 ,$$
(5)

where $x$ is the coordinate aligned with the mean wind direction, $y$ is the vertical
direction, $\kappa$ is the von Karman constant, and $F_x$ is the force per unit volume that the
snow particles exert on the fluid in the stream-wise direction and can be expressed as
$$F_x = \sum_{i=1}^{n} m_p a_i .$$
(6)

where $n$ is the number of particles per unit volume of fluid at height $y$, and $a_i$ is the
horizontal acceleration of particle $i$.





When the bed shear stress is greater than the threshold value, snow particles begin
lifting off the surface. The number of aerodynamically entrained snow particles $N_a$ is
(Shao and Li, 1999)

$$N_a = \varsigma u_* \left(1 - \frac{u_{*_t}^2}{u_*^2}\right) D^{-3} .\qquad(7)$$

where $\varsigma$ is a dimensionless coefficient ($1 \times 10^{-3}$ in our simulations), $u_*$ is the friction
velocity, and $u_{*_t}$ is the threshold friction velocity. Following the previous saltation
models (McEwan and Willetts, 1993), the vertical speed of all aerodynamically
entrained particles is $\sqrt{2gD}$ .

9         The following three splash functions for drifting snow proposed by Sugiura and

Maeno (2000) based on experiments are used to determine the number and motion
state of the splashed particles.

$$S_v(e_v) = \frac{1}{\beta^\alpha \Gamma(\alpha)} e_v^{\alpha-1} \exp\left(-\frac{e_v}{\beta}\right),\qquad(8)$$

$$S_h(e_h) = \frac{1}{\sqrt{2\pi\sigma^2}} \exp\left[-\frac{(e_h-\mu)^2}{2\sigma^2}\right],\qquad(9)$$

$$S_e(n_e) = {}_mC_{n_e} p^{n_e} (1-p)^{m-n_e} .\qquad(10)$$

In Eq. (8), $S_v$ is the probability distribution of the vertical restitution coefficient $e_v$,
$\Gamma(\alpha)$ is the gamma function, and $\alpha$ and $\beta$ are the shape and scale parameters for the
gamma distribution function. In Eq.(9), $S_h$ is the probability distribution of the
horizontal restitution coefficient $e_h$, and $\mu$ and $\sigma$ are the mean and variance,
respectively. In Eq. (10), $S_e$ is the probability distribution function of the number of
ejected particles $n_e$, a binomial distribution function with the mean $mp$ and the
variance $mp(1-p)$.

22        The potential temperature $\theta$ and specific humidity $q$ of the ambient air satisfy the

following prognostic equations (Déry and Yau, 1999)





$$\frac{\partial \theta}{\partial t} = \frac{\partial}{\partial y}\left( K_\theta \frac{\partial \theta}{\partial y} \right) - \frac{L_s S}{\rho_f C}, \qquad (11)$$

$$\frac{\partial q}{\partial t} = \frac{\partial}{\partial y}\left( K_q \frac{\partial q}{\partial y} \right) + \frac{S}{\rho_f} - Q. \qquad (12)$$

where $K_\theta = \kappa u_* y + K_T$ and $K_q = \kappa u_* y + K_V$ are the heat and moisture diffusivities (the
sum of eddy diffusivity and molecular diffusivity), respectively, $S$ is the sublimation
rate summed over all particles at each height above the surface, $L_s$ is the latent heat of
sublimation ($2.835 \times 10^6$ J kg$^{-1}$), $C$ is the specific heat of air, $Q = u \frac{\partial q}{\partial x}$ is the horizontal
advection of moisture at each height above the surface, and $\frac{\partial q}{\partial x}$ represents the
horizontal gradient in specific humidity. When the external dry air with specific
humidity $q_{out}$ enters into the study domain, we hypothesize that the specific humidity
in the study domain is linearly distributed along the horizontal direction and
possesses the value of $q_{in}$ at the exit. Thus, the horizontal advection of moisture can be
simplified to $Q = u(q_{in} - q_{out})/l$, with $l$ being the length of the domain.
The total DSS rate $Q_S$ (kg s$^{-1}$) of the saltation layer within the computational
domain is obtained by summing the mass loss of all saltating particles in the domain.

$$Q_S = \sum_i \left( \frac{dm}{dt} \right)_i, \qquad (13)$$

where $\left( \frac{dm}{dt} \right)_i$ is the mass loss rate corresponding to the $i$-th particle. At the air
temperature $T$ and undersaturation $\delta \ (= 1 - RH)$, the rate of mass change of a single
particle with radius $r$ due to sublimation is (Thorpe and Mason, 1966)
$$\frac{dm}{dt} = \frac{2\pi r \delta}{\frac{L_s}{KTNu}(\frac{L_s}{R_v T} - 1) + \frac{R_v T}{D_v She_s}}, \qquad (14)$$

where $RH$ is the relative humidity of air, $K$ is the molecular thermal conductivity of
the atmosphere ($0.024$ J m$^{-1}$ s$^{-1}$ K$^{-1}$), $D_v$ is the molecular diffusivity of water vapor in
the atmosphere, $R_v$ is the gas constant for water vapor ($461.5$ J kg$^{-1}$ K$^{-1}$), $e_s$ is saturated





vapor pressure with respect to an ice surface, and $Nu$ and $Sh$ are the Nusselt number
and the Sherwood number, respectively, both of which are dimensionless and depend
on the wind velocity and particle size (Thorpe and Mason, 1966; Lee, 1975).

$$Nu = Sh = \begin{cases} 1.79 + 0.606\,\mathrm{Re}^{0.5} & 0.7 < \mathrm{Re} < 10 \\ 1.88 + 0.580\,\mathrm{Re}^{0.5} & 10 < \mathrm{Re} < 200 \end{cases}. \qquad (15)$$

where $\mathrm{Re} = DV_r / \upsilon$ is the Reynolds number and $\upsilon$ is the kinematic viscosity of air.
For the purpose of comparison with the sublimation of suspended particles, the
initial relative humidity profile in accordance with that of Xiao et al. (2000) is

$$RH = 1 - R_S \ln(y / y_0), \qquad (16)$$

where $y_0$ is roughness length and $R_S = 0.039469$.
The conversion relation between relative humidity and specific humidity is

$$q = 0.622 \cdot \frac{e_s}{p - e_s} \cdot RH, \qquad (17)$$

where $e_s = 610.78 \exp\left[21.87(T - 273.16)/(T - 7.66)\right]$.
The constant initial potential temperature $\theta_0$ is 263.15K (but is 253.16 K in the
comparison with Xiao et al. (2000)) and the initial absolute temperature is

$$T_0 = \theta_0 \left(\frac{p}{p_0}\right)^{0.286}, \qquad (18)$$

where $p$ is the pressure and its initial distribution is based on the hypsometric
equation

$$p = p_0 \exp\left(-\frac{yg}{R_d \theta_0}\right). \qquad (19)$$

where $p_0$ is taken as 1000 hPa and $R_d$ is the gas constant for dry air (287.0 J kg$^{-1}$ K$^{-1}$).
**2.2  Calculation Procedure**
The procedure for the calculations is enumerated below.





1. The length, width and height of the computational domain sampled from the saltation layer above the surface are 1.0 m, 0.01 m, and 1.0 m, respectively. The initial and boundary conditions of temperature and humidity are set from Eqs. (16)-(19).
2. Snow particles are considered as spheres with diameter of 200 $\mu m$ and density of 910 kg m$^{-3}$. The threshold friction velocity of snow is 0.21 m s$^{-1}$ and the snow bed roughness is $3.0 \times 10^{-5}$ m (Nemoto and Nishimura, 2001).
3. The initial wind field is logarithmic. If the bed shear stress is greater than the threshold value, particles are entrained from their random positions on the snow surface at vertical speed $\sqrt{2gD}$ and the number of aerodynamically entrained snow particles satisfies Eq. (7).
4. The snow particle trajectory is calculated using Eqs. (1)-(4) every 0.00001 s in order to obtain the velocity used in the calculation of sublimation rate and the new location of each drifting snow particle to determine whether the snow particle falls on the snow bed.
5. As the snow particles fall on the snow bed, where they impart their energy to other snow particles and splash or eject other snow particles, the velocity and angle of the ejected particles satisfy the splash functions, i.e., Eqs. (8)-(10), according to the motion state of the incident particles and the actual wind field at that time. The number of snow particles is re-counted every 0.00001 s.

21    6. The reactive force $F_x$ that the snow particles exert on the wind field induces wind modification according to Eq. (5).

23    7. Based on the process above, the velocity and location of each drifting snow particle are derived and then used in Eqs. (13)-(15) to calculate their sublimation rate every 0.00001 s. Under the effect of DSS, potential temperature and specific humidity at different heights under the diffusion or advection moisture transport are calculated every 0.00001 s.

28    8. The new values of wind field calculated in step 6 are used in step 3, and then steps 4 to 7 are recalculated. Such a cycle is repeated to finish the calculation of DSS under thermodynamic effects. Each calculation takes 60 s.

31    **3  Results and Discussion**



## 3.1 Relative Humidity and Temperature

The relative humidity at 1 cm height for different defined wind velocities generally
reaches saturation within 10 s when moisture transport is not included (Fig. 1a).
Snow sublimation will not occur, and the temperature will not change (Fig. 1b).
However, when moisture transport is included, the snow sublimation occurs
throughout the simulation period, and temperature decreases continually. Moreover,
under the same moisture transport mechanism, the greater the wind friction velocity,
the higher the relative humidity and temperature change (Fig. 1). The relative
humidity at 1 cm shows a trend of rapid decrease, then rapid increase, and finally a
slow increase (Fig. 1a), but does not reach saturation in the simulation period of 60 s.
Early in the wind-blown snow stage, the sublimation rate is smaller as only a few
saltating particles sublime and the moisture at the lower height largely moves
outwards due to the effect of moisture transport, resulting in relative humidity
decrease. With continuing wind-blown snow, more snow particles leave the surface,
which increases the sublimation rate and hence the relative humidity. When it reaches
a steady state, the amount of snow particles in the saltation layer will no longer
increase, but fluctuate within a certain range. Thereafter, because of the increase in
humidity and cooling, DSS weakens (Fig. 2). The results indicate that DSS in the
saltation layer has a self-limiting nature.

## 3.2 Sublimation Rate

Moisture transport could remove some moisture, attenuating the increase of relative
humidity and thus negative feedback, leading to higher sublimation rates with
moisture transport than without (Fig. 2). With moisture removal only by diffusion,
the sublimation rate at 60 s is roughly the same at 3 wind velocities, meaning that
sublimation still shows obvious negative feedback. However, with moisture transport
by diffusion and advection, the sublimation rate increases significantly as the
negative feedback effect is effectively reduced. Moreover, the sublimation rate
increases with the friction velocity and can be even greater than that at the highest
wind velocity without advection. For example, the sublimation rate at 60 s with
advection is $0.61 \times 10^{-5}$ kg m$^{-2}$ s$^{-1}$ at a friction velocity of 0.3 m s$^{-1}$, greater than that of
$0.44 \times 10^{-5}$ kg m$^{-2}$ s$^{-1}$ at a friction velocity of 0.5 m s$^{-1}$ without considering advection.
The sublimation rate even reaches $0.96 \times 10^{-5}$ kg m$^{-2}$ s$^{-1}$, equaling the 0.83 mm d$^{-1}$





snow water equivalent (SWE) at a friction velocity of 0.5 m s$^{-1}$ with advection
included (Fig. 2). Furthermore, sublimation continues to occur. Thus, it can be seen
that effective moisture transport can weaken the negative feedback of sublimation,
hence significantly affecting DSS. Because the occurrence of wind-blown snow must
coincide with the airflow, DSS in the saltation layer is not negligible, and the
assumption that the saltation layer is a saturation boundary layer is inadvisable.

7       Air temperature decreases with decreasing height, along with air saturation degree

during wind-blown snow, which is adverse to sublimation in contrast to higher
heights above the surface. Nevertheless, the volume sublimation rate increases with
decreasing height (Fig. 3). This is in agreement with the vertical profiles of the
horizontal mass flux of snow particles (Huang et al., 2011). That is, there are more
snow particles that can participate in sublimation at lower heights, leading to higher
sublimation rates even in environments adverse to sublimation. The results indicate
that the particle density is an important controlling factor for sublimation rate, which
is consistent with a previous study (Wever et al., 2009). A comparison between our
simulated results and that of four models for suspended snow, i.e., PIEKTUK-T,
WINDBLAST, SNOWSTORM and PIEKTUK-B, shows that the local sublimation
rate of the suspended snow at 60 s can reach $10^{-6}$ kg m$^{-3}$ s$^{-1}$ at most (Xiao et al.,2000)
(Fig. 3), smaller than that of our calculated results ($10^{-4}$ kg m$^{-3}$ s$^{-1}$) by 2 orders of
magnitude at the same initial temperature and relative humidity. This result shows
that the assumption that sublimation in the saltation layer can be ignored by
considering it a saturation boundary layer is inadvisable. Therefore, DSS in the
saltation layer is of non-negligible importance and requires further detailed study.
**4 Conclusions**
In this study, we established a wind-blown snow model and balance equations for
heat and moisture to study the effect of different moisture transport mechanisms on
DSS in the saltation layer. As has been reported (e.g., Schmidt, 1982), DSS could
lead to strong increases in humidity and cooling, which in turn can significantly
reduce the DSS rate, i.e., DSS has an inherently self-limiting nature. Moreover, the
relative humidity in the saltation layer quickly reaches saturation when moisture
transport is not considered. However, effective moisture transport, such as advection,
can dramatically weaken the negative feedback of sublimation and prolong the
duration of the higher DSS rate and hence has a profound effect on DSS. Because of




the presence of advection, DSS rate increases with the friction velocity and the
volume sublimation rate of saltating particles is several orders of magnitude greater
than that of the suspended particles due to the higher particle density in the saltation
layer. Thus, DSS in the saltation layer plays an important part in the energy and mass
balance of snow cover and needs to be further studied.
**Acknowledgments**
This work is supported by the State Key Program of National Natural Science
Foundation of China (91325203), the National Natural Science Foundation of China
(41371034), and Innovative Research Group of the National Natural Science
Foundation of China (11421062).





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



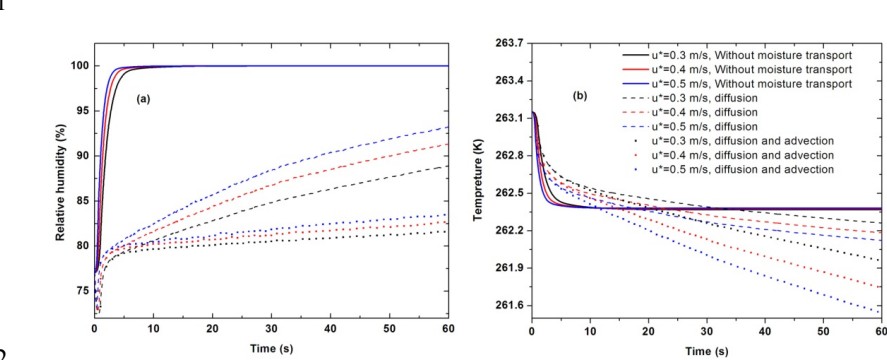

Figure 1.Temporal evolution of relative humidity (a) and temperature (b) at 1 cm
above the surface for three wind force levels neglecting the effects of moisture
transport, considering only moisture diffusion, and both moisture diffusion and
advection.



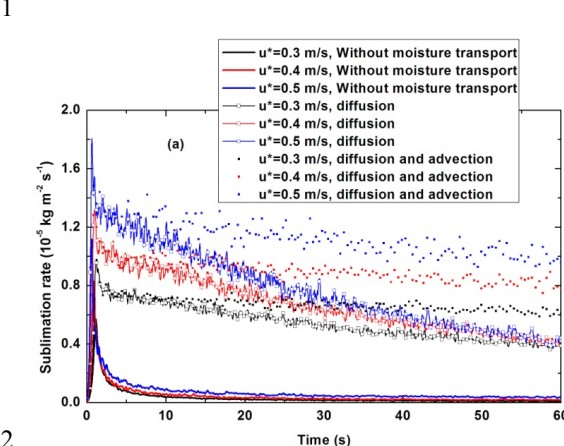

3 Figure 2.Temporal evolution of drifting snow sublimation rate for three wind force

4 levels neglecting moisture transport, considering only moisture diffusion, and both

5 moisture diffusion and advection.



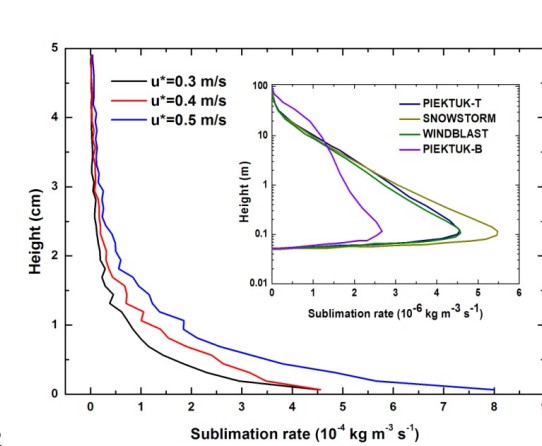

Figure 3. Comparison of the sublimation rate for the saltation layer and suspension
layer (the inset figure) at 60 s as a function of height. The inset figure shows the
sublimation rate of four models for the suspension layer with initial friction velocity
of 0.87 m s$^{-1}$ reported in Xiao et al. (2000). Our results for the sublimation rate in the
saltation layer are obtained for three wind force levels (<0.87 m s$^{-1}$) with moisture
diffusion and advection included with the same initial temperature (253.16 K) and
relative humidity as Xiao et al. (2000).