# Peer review of "The impacts of moisture transport on drifting snow sublimation in the saltation layer"

_Atmospheric Chemistry and Physics, 2015_

## Referee Comment (RC1) · Anonymous Referee #1 · 8 Mar 2016

This contribution addresses an important problem, which is globally unresolved namely how much snow mass is lost back to the atmosphere during drifting and blowing snow. The contribution now tries to quantify snow sublimation in the saltation layer, which is typically regarded as insignificant because of quick saturation. The authors present a concise and well-written article, which nicely discusses her main hypothesis that continuous transport of moisture out of the saltation layer may play a significant role. However, their model assessment is fundamentally flawed and the paper must therefore be rejected. In the quantitative model assessment, the authors introduce (p.7 l.11) a completely arbitrary sink of moisture due to "advective" transport, which is contradicting the model set-up as a boundary layer model. The rest of the model uses the assumption of an equilibrium boundary layer in which forces or sinks/sources are balanced by vertical turbulent transport. By superimposing an artificial and completely unjustified

horizontal moisture transport term, you can produce any number for sublimation. The calculation results are therefore not a scientifically sound basis for the conclusion that "DSS rate in the saltation layer can be several orders of magnitude greater than that of the suspended particles".

There are a (small) number of minor comments such as a missing discussion on surface sublimation or a splash function description on p.9 l.18, which appears to not match the corresponding equations but these are not important compared to the erroneous model set up described above.

---

## Short Comment (SC1) · 22 Mar 2016

In this study, a 2-D snow drift model is introduced. A saltation model is coupled with a treatment of moisture and temperature changes associated with snow drift. Much emphasis is placed on the description of the model, and it is relatively short on discussion of the model results. The paper is reasonably well written. A somewhat disappoiting aspect of the paper is that it does not have measurements for comparison or even published data for comparison. May be this can be improved.

The self-limiting nature of snow drift process is clearly revealed. This self-limiting process is similar to saltation of sand with no sublimation, but appears to be more complex, as it involves the moisture process. It is not clear however how the modified stability of the flow influences the self-limiting process.

[Figure]

The authors did not consider the effect of turbulence. While including turbulence may be more difficult, the authors may wish to discuss what might happen if turbulence is included. This is also important, because the stability of the saltation layer also affects the profile of the mean wind. Indeed, I do not see where thermal stability is included in the model.

There are minor writing problems which the authors should carefully check again, for example, DSS is not defined.
* * *

---

## Referee Comment (RC3) · S. Simoëns (Referee) · 25 Mar 2016

In this study the authors used a 2D model for studying snow sublimation in regards to its capacity to impact saltation layer evolution. For this aim they used a solid particle transport modeling combined with transport equations for potential temperature and specific humidity. The modeling mimics splash and take-off processes as usual. The two way coupling between sublimation and velocity is acted via a rough term inside longitudinal velocity field evolution equation. No four way coupling is introduced. The paper described clearly how all processes are accounted for.

Some important points are the following :

The two way coupling has to be more deeply discussed as it is proposed : what are the hypothesis leading to this formulation ? Furthermore even if variation of temperature

are low it could be discussed if it could induced some effects on velocity field particularly where there is a large concentration of snow particles. It is relatively easy to add in such modeling these effects. Concerning the four way coupling and contrarily to sand it could probably modifies significantly the budget close to the ground. As writen by the authors such modeling is not time consuming in so it would be interesting to time increase the computational runs to observe if the various time evolutions of the shown quantities are stabilised.

Concerning results it would be interesting to plot snow particles concentration profile evolutions. I agree that it would be interesting to account for turbulence and to compare with experimental results.

could you check equation (14) (or may be I did a mistake but molecular weight of water have to be taken into account in the fisrt term of the denominator) ? Some details concerning some threshold or constant as the one for take-off have to be given as they are crucial.

However the aim of this work is interesting and also the way to treat it.

---

## Referee Comment (RC4) · Anonymous Referee #3 · 12 Apr 2016

In this study, the authors present an interesting research to evaluate the effect of drafting snow sublimation in the saltation layer, which is rarely studied, but important for the hydrological balance of snow cover. A snowdrift model with considering the coupling effects of snow sublimation, temperature, humidity and moisture transport is established to address their ideas and research. They describe their models and methods concisely and clearly and analyze the results reasonably. They present a well-written article, but the Results section seems a little short and has no comparison with the measurements or published data.

The authors need to carefully check their manuscript again to do a better job of defining the parameters they use for the controlling equations, for example, the definition of the sublimation rate S.

[Figure]

The prognostic equations of potential temperature and specific humidity (Eqs. 11 and 12) seem to be different from the reference, which may lead to misunderstandings (just as the comments of reviewer 1). In order to express the calculation of temperature and humidity more clearly, it would be better to derive the prognostic equations from the basic convection diffusion equation.

Why each calculation takes 60 s and doesn't show the sublimation rate when it tends to be stabilized?

I agree that it could be discussed if the variation of temperature could induce some effects on velocity field. The evolutions of snow particles concentration should be discussed.

On the whole, this work is a fundamental research and obviously focusing on science common topics, which is interesting and well present.

---

## Author Comment (AC1) · 25 Apr 2016

We'd like to thank Referee 1 for the comments. We have studied the comments carefully and the responds are as follows.

This contribution addresses an important problem, which is globally unresolved namely how much snow mass is lost back to the atmosphere during drifting and blowing snow. The contribution now tries to quantify snow sublimation in the saltation layer, which is typically regarded as insignificant because of quick saturation. The authors present a concise and well-written article, which nicely discusses her main hypothesis that continuous transport of moisture out of the saltation layer may play a significant role.

Reply: Thanks for the reviewer's recognition of the importance and the writing of our manuscript.

[Figure]

However, their model assessment is fundamentally flawed and the paper must therefore be rejected. In the quantitative model assessment, the authors introduce (p.7 l.11) a completely arbitrary sink of moisture due to "advective" transport, which is contradicting the model set-up as a boundary layer model. The rest of the model uses the assumption of an equilibrium boundary layer in which forces or sinks/sources are balanced by vertical turbulent transport. By superimposing an artificial and completely unjustified horizontal moisture transport term, you can produce any number for sublimation. The calculation results are therefore not a scientifically sound basis for the conclusion that "DSS rate in the saltation layer can be several orders of magnitude greater than that of the suspended particles".

Reply: As we know, the term (i.e. Q in our paper) described the advection of mean total moisture by the mean wind basically exists in the moisture conservation equation (Roland B. Stull, 1988). Normally, this term is ignored under the hypothesis of horizontal homogeneous, which leads to the condition mentioned by the referee "an equilibrium boundary layer in which forces or sinks/sources are balanced by vertical turbulent transport". Although it is an effective way to simplify the moisture conservation equation for an infinite planar snow cover, this hypothesis is sometimes hard to be satisfied because of some common phenomena, such as patchy mosaic of snow or heterogeneous snow drifting over rough surface (Liston, 1999). In the paper, we did not artificially add the horizontal moisture transport term, but took into account of the ignored term under more complex situation. Actually, the effect of advection is considered via the setups of the entrance boundary and the egress boundary of moisture for the simulated region in our paper. Two extreme conditions were discussed: qin=qout , which represents the advection was not took into account and the other case qin represents moisture of dry air and qout equals to the moisture of air affected by the snowdrift sublimation in the simulated region. Although these two cases may not correspond to the really situation exactly, it is an acceptable and useful way to discuss the possible influence of horizontal advection. Similar method was employed by other researchers, such as Richard Bintanja (2001). Anyway we honestly appreciate the referee for the

suggestions and we also found there are some irrelevant descriptions which may cause misunderstanding in our paper. We will carefully check the presentation (including relevant equations) of this part and improve it in the revised version.

There are a (small) number of minor comments such as a missing discussion on surface sublimation or a splash function description on p.9 l.18, which appears to not match the corresponding equations but these are not important compared to the erroneous model set up described above.

Reply: In this manuscript the initial relative humidity profile is set as equation 16 , therefore there will be a saturated layer near the surface for the surface sublimation. As for the description of the splash function and other writing problems, we will try our best to check our manuscript carefully again and improve it in the revised version.
* * *

---

## Author Comment (AC3) · 29 Apr 2016

We'd like to thank Referee #3 for the insightful comments and positive evaluation of our work. We have studied comments carefully and will do our best to revise and improve our manuscript. The comments by the reviewer are repeated and the responds are as follows.

In this study, the authors present an interesting research to evaluate the effect of drafting snow sublimation in the saltation layer, which is rarely studied, but important for the hydrological balance of snow cover. A snowdrift model with considering the coupling effects of snow sublimation, temperature, humidity and moisture transport is established to address their ideas and research. They describe their models and methods concisely and clearly and analyze the results reasonably.

Reply: We thank the reviewer for the positive evaluation of our work.

They present a well-written article, but the Results section seems a little short and has no comparison with the measurements or published data.

Reply: Just like the reviewers and Prof. Yaping Shao, we also think that some measurements are necessary for validation of our simulation results. Unfortunately, the measurement of snowdrift sublimation in the saltation layer is very difficult to conduct at the present stage. Even so, we will try to conduct such measurements in our future studies.

The authors need to carefully check their manuscript again to do a better job of defining the parameters they use for the controlling equations, for example, the definition of the sublimation rate S.

Reply: Thanks. We will carefully check the manuscript again to make the parameters of the controlling equations more clearly.

The prognostic equations of potential temperature and specific humidity (Eqs. 11 and 12) seem to be different from the reference, which may lead to misunderstandings (just as the comments of reviewer 1). In order to express the calculation of temperature and humidity more clearly, it would be better to derive the prognostic equations from the basic convection diffusion equation.

Reply: Thanks for the insightful comment. We will derive the prognostic equations of potential temperature and specific humidity from the basic convection diffusion equation in the revised manuscript.

Why each calculation takes 60 s and doesn't show the sublimation rate when it tends to be stabilized?

Reply: Thanks for the insightful comment. By considering of the required time of drifting snow development and the capability of computer, the simulated time was set as 60s, which is significantly surpass drifting snow development time (about 2-3 s) and

could be actualized easily on PC. That is indeed that, the sublimation rate is hard to be stabilized in this 60s, but our results could expose the issues that we care about. Theoretically, the snow sublimation may vanish finally with the development of time. As we state in our manuscript, DSS has an inherent self-limiting nature due to the feedback associated with the heat and moisture budgets. On one hand, snow sublimation absorbs heat, which decreases the temperature of the ambient air and the saturation vapor pressure; on the other hand, it will induce the increment in the moisture content of the ambient air. Both of the above two points can increase the relative humidity of ambient air. It may lead to a saturated layer near the surface finally, and thus sublimation will vanish. Here, we show the temporal evolution of drifting snow sublimation within 60 s to compare our results with the previous study.

I agree that it could be discussed if the variation of temperature could induce some effects on velocity field. The evolutions of snow particles concentration should be discussed.

Reply: Thanks. Indeed, the variation of temperature could induce some effects on velocity field. However, we found that this effect can be ignored by testing (Fig. 1). In our study, the temperature varies between 261.5 K and 263.15 K. The variation of temperature due to snow sublimation is below 2 K, which is relatively low. From Figure 1, we can see that the effect of variation of temperature on velocity field is very small. Thus, we didn't take this effect into consideration. In addition, following the reviewer's suggestion, we will add a figure to show snow particles concentration profile evolution in the revised manuscript.

On the whole, this work is a fundamental research and obviously focusing on science common topics, which is interesting and well present.

Reply: Thanks for the positive evaluation of our work.

[Figure]

[Figure]

Fig. 1.

[Figure]

---

## Author Comment (AC4) · 29 Apr 2016

We'd like to thank Referee #2 for the insightful comments and positive evaluation of our work. We have studied comments carefully and will do our best to revise and improve our manuscript. The comments by the reviewer are repeated and the responds are as follows.

In this study the authors used a 2D model for studying snow sublimation in regards to its capacity to impact saltation layer evolution. For this aim they used a solid particle transport modeling combined with transport equations for potential temperature and specific humidity. The modeling mimics splash and take-off processes as usual. The two way coupling between sublimation and velocity is acted via a rough term inside longitudinal velocity field evolution equation. No four way coupling is introduced. The

[Figure]

paper described clearly how all processes are accounted for.

Reply: We thank the reviewer for the positive evaluation of our work. In this study, a wind-blown snow model that takes into consideration of the coupling effect between wind and snow particles is established to simulate the saltating process of snow particles. Then balance equations for heat and moisture of an atmospheric boundary layer, and an equation for the rate of mass loss of a single ice sphere due to sublimation were combined to study the sublimation rate of drifting snow by tracking each saltating particle in drifting snow. The splash functions for drifting snow used in this manuscript was proposed by Sugiura and Maeno (2000) based on their experiments, which is used to determine the number and motion state of the splashed particles as usual.

Some important points are the following:

The two way coupling has to be more deeply discussed as it is proposed: what are the hypothesis leading to this formulation?

Reply: Thanks. We will modify our manuscript to make our model clearer.

Furthermore even if variation of temperature are low it could be discussed if it could induced some effects on velocity field particularly where there is a large concentration of snow particles. It is relatively easy to add in such modeling these effects.

Reply: Thanks for the insightful comment. The reviewer is right. The variation of temperature could induce some effects on velocity field. However, we found that this effect can be ignored by testing (Fig. 1). In our study, the temperature varies between 261.5 K and 263.15 K. The variation of temperature due to snow sublimation is below 2 K, which is relatively low. From Figure 1 we can see that the effect of variation of temperature on velocity field is very small. Thus, we didn't take this effect into consideration.

Concerning the four way coupling and contrarily to sand it could probably modifies significantly the budget close to the ground. As written by the authors such modeling

is not time consuming in so it would be interesting to time increase the computational runs to observe if the various time evolutions of the shown quantities are stabilised.

Reply: Thanks. As we state in our manuscript, DSS has an inherent self-limiting nature due to the feedback associated with the heat and moisture budgets. On one hand, snow sublimation absorbs heat, which decreases the temperature of the ambient air and the saturation vapor pressure; on the other hand, it will induce the increment in the moisture content of the ambient air. Both of the above two points can increase the relative humidity of ambient air. It may lead to a saturated layer near the surface finally, and thus sublimation may vanish.

Concerning results it would be interesting to plot snow particles concentration profile evolutions.

Reply: Thanks. Following the reviewer's suggestion, we will add a figure to show snow particles concentration profile evolution in the revised manuscript.

I agree that it would be interesting to account for turbulence and to compare with experimental results.

Reply: Thanks for the insightful comment. We acknowledge the comment that some studies (Jasper F. Kok and Nilton O. Renno, 2009; Yaping Shao, 2010) did include the effects of turbulent in their saltation model and it was found that turbulent flow substantially affects the saltation movement of sand particles, mainly the movement of small particles. But the effect of turbulence on larger saltating particles is much less pronounced for their larger inertia and thus smaller susceptibility to fluid velocity perturbations. For example, Shao (2010) showed the effect of turbulence flow for 200 micrometers particles in a logarithmically-profiled airflow (u*=0.5m/s) is small. In our simulations, the diameter of the snow particles is 200 micrometers. Furthermore, this study concentrates on the time-averaged contributions of saltating snow particles to snow sublimation. Therefore we didn't take into consideration of the effect of turbulence in this manuscript. Perhaps we need to include turbulent effects in our future work. Just

like the reviewer and Prof. Yaping Shao, we also think that some measurements are necessary for validation of our simulation results. Unfortunately, the measurement of snowdrift sublimation in the saltation layer is very difficult to conduct at the present stage. Even so, we will try to conduct such measurements in our future studies.

could you check equation (14) (or may be I did a mistake but molecular weight of water have to be taken into account in the first term of the denominator) ? Some details concerning some threshold or constant as the one for take-off have to be given as they are crucial.

Reply: Thanks. Indeed, the first term of the denominator in the sublimation rate given by Thorpe and Mason is related to the molecular weight of water M (kg/mol), as well the universal gas constant R (J/mol/K). But in our study, the gas constant for water vapor Rv (=R/M) is defined with the value of 461.5 (J/kg/K). Following the reviewer's suggestions, we will describe these parameters in detail to make them clearly in the revised manuscript.

However the aim of this work is interesting and also the way to treat it.

Reply: Thanks for the positive evaluation of our work.
* * *
[Figure]

Fig. 1.

---

## Author Response (AR1)

We'd like to thank Referees for the insightful comments and positive evaluation of our work. We have studied the comments carefully and did our best to revise and improve our manuscript. The comments by the reviewer are repeated in bold font and the responds are as follows.

*Referee #1*

**This contribution addresses an important problem, which is globally unresolved namely how much snow mass is lost back to the atmosphere during drifting and blowing snow. The contribution now tries to quantify snow sublimation in the saltation layer, which is typically regarded as insignificant because of quick saturation. The authors present a concise and well-written article, which nicely discusses her main hypothesis that continuous transport of moisture out of the saltation layer may play a significant role.**

**Reply**: Thanks for the reviewer's recognition of the importance and the writing of our manuscript.

**However, their model assessment is fundamentally flawed and the paper must therefore be rejected. In the quantitative model assessment, the authors introduce (p.7 l.11) a completely arbitrary sink of moisture due to "advective" transport, which is contradicting the model set-up as a boundary layer model. The rest of the model uses the assumption of an equilibrium boundary layer in which forces or sinks/sources are balanced by vertical turbulent transport. By superimposing an artificial and completely unjustified horizontal moisture transport term, you can produce any number for sublimation. The calculation results are therefore not a scientifically sound basis for the conclusion that "DSS rate in the saltation layer can be several orders of magnitude greater than that of the suspended particles".**

**Reply**: As we know,the term (i.e. Q in our paper) described the advection of total moisture by the wind basically exists in the moisture conservation equation (Roland B. Stull, 1988). Normally, this term is ignored under the hypothesis of horizontal homogeneity, which leads to the condition mentioned by the referee "an equilibrium boundary layer in which forces or sinks/sources are balanced by vertical turbulent transport". Although it is an effective way to simplify the moisture conservation equation for an infinite planar snow cover, this hypothesis is sometimes hard to be

satisfied because of some common phenomena, such as patchy mosaic of snow or heterogeneous snow drifting over rough surface (Liston, 1999). In the paper, we did not artificially add the horizontal moisture transport term, but took into account of the ignored term under more complex situation. Actually, the effect of advection is considered via the setups of the entrance boundary and the egress boundary of moisture for the simulated region in our paper. Two typical conditions were discussed: 1), $q_{in}=q_{out}$ , which neglects the effect of advection and be corresponding to condition of infinite and homogeneous snow cover; 2), to consider advection effect at the edge of snow surface, $q_{in}$ represents moisture of dry air and $q_{out}$ equals to the moisture of air affected by the snowdrift sublimation in the simulated region. Although these two cases may not correspond to the really situation exactly, it is an acceptable and useful way to discuss the possible influence of horizontal advection. Similar method was employed by other researchers, such as Richard Bintanja (2001). Anyway we honestly appreciate the referee for the suggestions and we also found there are some irrelevant descriptions which may cause misunderstanding in our paper. Relevant modifications are shown as following:

We described two extreme cases, 1) neglecting the effects of moisture transport; 2) considering moisture transport due to both moisture diffusion and advection. And also a common situation, i.e. considering only moisture diffusion, was studied to explore the interaction between snow sublimation and moisture transport in the revised manuscript. Moreover, we derived the prognostic equations of potential temperature and specific humidity (Eqs. (13) and (14)) from the basic convection diffusion equation (Eqs. (11) and (12)).

**There are a (small) number of minor comments such as a missing discussion on surface sublimation or a splash function description on p.9 l.18, which appears to not match the corresponding equations but these are not important compared to the erroneous model set up described above.**

Reply: In this manuscript, the initial relative humidity profile is set as $RH = 1 - R_S \ln(y / y_0)$ , therefore there will be a saturated layer adjacent the surface. In this case, surface sublimation will not occur. When moisture transport is considered, it is still nearly saturated. Thus, we didn't take surface sublimation into consideration. For splash function, some improvements of description are made here, including: the definition of the vertical restitution coefficient $e_v$ and the horizontal

restitution coefficient $e_h$ .

*Short comment #1*

**In this study, a 2-D snow drift model is introduced. A saltation model is coupled with a treatment of moisture and temperature changes associated with snow drift. Much emphasis is placed on the description of the model, and it is relatively short on discussion of the model results. The paper is reasonably well written. A somewhat disappoiting aspect of the paper is that it does not have measurements for comparison or even published data for comparison. May be this can be improved.**

**Reply**: Thanks for the insightful comments and positive evaluation of our work. As mentioned in our manuscript, previous studies on drifting snow sublimation mainly concentrated on the suspended snow. Whereas the sublimation of salating particles was generally ignored due to the consideration that sublimation will soon vanish in the saltation layer for the feedback of drifting snow sublimation (DSS) may lead to a saturated layer near the surface. Therefore, there are very few existing studies and published data on snowdrift sublimation in the saltation layer. In this manuscript, we only give a comparison of snowdrift sublimation between saltating and suspended snow to clarify the importance of drifting snow sublimation in the saltation layer.

Just like Prof. Yaping Shao, we also think that some measurements are necessary for validation of our simulation results. However, the measurement of snowdrift sublimation in the saltation layer is very difficult to conduct at the present stage. We have tried our best to make our results comparison with the published studies. Unfortunately, suitable data for comparison were not found.

**The self-limiting nature of snow drift process is clearly revealed. This self-limiting process is similar to saltation of sand with no sublimation, but appears to be more complex, as it involves the moisture process. It is not clear however how the modified stability of the flow influences the self-limiting process.**

**Reply**: Thanks for the insightful comments. In this study, a wind-blown snow model, balance equations for heat and moisture of an atmospheric boundary layer, and an equation for the rate of mass loss of a single ice sphere due to sublimation were combined to study the sublimation rate of

drifting snow by tracking each saltating particle in drifting snow. Therefore, the influence of flow on the snow sublimation is mainly evaluated by influencing the processes of saltation movement (equation 5) and the horizontal advection of moisture (equation 12). Because the maximum mass loss of the sublimation for a single snow particle is less than one thousandth of the particle' mass during a process of saltation movement under the conditions of this study, we didn't evaluate mass change of the particle on the air flow.

**The authors did not consider the effect of turbulence. While including turbulence may be more difficult, the authors may wish to discuss what might happen if turbulence is included. This is also important, because the stability of the saltation layer also affects the profile of the mean wind. Indeed, I do not see where thermal stability is included in the model.**

**Reply**: Thanks. We acknowledge the comment that some studies (Jasper F. Kok and Nilton O. Renno, 2009; Yaping Shao, 2010) did include the effects of turbulent in their saltation model and it was found that turbulent flow substantially affects the saltation movement of sand particles, mainly the movement of small particles. But the effect of turbulence on larger saltating particles is much less pronounced for their larger inertia and thus smaller susceptibility to fluid velocity perturbations. For example, Shao (2010) showed the effect of turbulence flow for 200 μm particles in a logarithmically-profiled airflow ($u_* = 0.5$ m/s) is small. In our simulations, the diameter of the snow particles is 200 μm. Furthermore, this study concentrates on the time-averaged contributions of saltating snow particles to snow sublimation. Therefore we didn't take into consideration of the effect of turbulence in this manuscript. Perhaps we need to include turbulent effects in our future work.

**There are minor writing problems which the authors should carefully check again, for example, DSS is not defined.**

**Reply**: Thanks for the comment. We have modified the writing problems. Following the reviewer's comment, we have defined DSS as drifting snow sublimation in the first sentence of the abstract in the revised manuscript.

*Referee #2*

**In this study the authors used a 2D model for studying snow sublimation in regards to its**

**capacity to impact saltation layer evolution. For this aim they used a solid particle transport modeling combined with transport equations for potential temperature and specific humidity. The modeling mimics splash and take-off processes as usual. The two way coupling between sublimation and velocity is acted via a rough term inside longitudinal velocity field evolution equation. No four way coupling is introduced. The paper described clearly how all processes are accounted for.**

**Reply**: We thank the reviewer for the positive evaluation of our work. In this study, a wind-blown snow model that takes into consideration of the coupling effect between wind and snow particles is established to simulate the saltating process of snow particles. Then balance equations for heat and moisture of an atmospheric boundary layer, and an equation for the rate of mass loss of a single ice sphere due to sublimation were combined to study the sublimation rate of drifting snow by tracking each saltating particle in drifting snow. The splash functions for drifting snow used in this manuscript was proposed by Sugiura and Maeno (2000) based on their experiments, which is used to determine the number and motion state of the splashed particles as usual.

**Some important points are the following:**

**The two way coupling has to be more deeply discussed as it is proposed: what are the hypothesis leading to this formulation?**

**Reply:** Thanks. In the manuscript, we have stated that the initial wind field is logarithmic and the mean horizontal wind velocity $u$ satisfies the Navier-Stokes equation. For a stable wind blowing over an infinite plane bed, according to Prandtl's mixing length theory we derived the coupling equation (5) to describe the interactive effect between the snow particles and wind field. For details, please see Lines 13-15 of Page 5 in the revised manuscript.

**Furthermore even if variation of temperature are low it could be discussed if it could induced some effects on velocity field particularly where there is a large concentration of snow particles. It is relatively easy to add in such modeling these effects.**

**Reply**: Thanks for the insightful comment. The reviewer is right. The variation of temperature could induce some effects on velocity field. However, we found that this effect can be ignored by testing (Fig. 1.1). In our study, the variation of temperature due to snow sublimation is below 2 K,

which is relatively low. From Figure 1.1 we can see that the effect of such variation of temperature on velocity field is very small. Thus, we didn't take this effect into consideration.

[Figure]

Figure 1.1   Wind velocity profile at different temperature

**Concerning the four way coupling and contrarily to sand it could probably modifies significantly the budget close to the ground. As writen by the authors such modeling is not time consuming in so it would be interesting to time increase the computational runs to observe if the various time evolutions of the shown quantities are stabilised.**

Reply: Thanks. From Fig. 3, we can see that DSS has reached steady state with moisture diffusion and advection considered. But when only moisture diffusion is included, it is hard to reached steady state. As we state in our manuscript, DSS has an inherent self-limiting nature due to the feedback associated with the heat and moisture budgets. On one hand, snow sublimation absorbs heat, which decreases the temperature of the ambient air and the saturation vapor pressure; on the other hand, it will induce the increment in the moisture content of the ambient air. Both of the above two points can increase the relative humidity of ambient air. It may lead to a saturated layer near the surface finally, and thus sublimation may vanish.

**Concerning results it would be interesting to plot snow particles concentration profile evolutions.**

Reply: Thanks. Following the reviewer's suggestion, we added a Figure to show the temporal evolution of snow transport rate and the profile of snow particle number density at the steady state

for three wind force levels in the revised manuscript.

**I agree that it would be interesting to account for turbulence and to compare with experimental results.**

**Reply**: Thanks for the insightful comment. We acknowledge the comment that some studies (Jasper F. Kok and Nilton O. Renno, 2009; Yaping Shao, 2010) did include the effects of turbulent in their saltation model and it was found that turbulent flow substantially affects the saltation movement of sand particles, mainly the movement of small particles. But the effect of turbulence on larger saltating particles is much less pronounced for their larger inertia and thus smaller susceptibility to fluid velocity perturbations. For example, Shao (2010) showed the effect of turbulence flow for 200 μm particles in a logarithmically-profiled airflow ($u_* = 0.5$ m/s) is small. In our simulations, the diameter of the snow particles is 200 μm. Furthermore, this study concentrates on the time-averaged contributions of saltating snow particles to snow sublimation. Therefore we didn't take into consideration of the effect of turbulence in this manuscript. Perhaps we need to include turbulent effects in our future work.

Just like the reviewer and Prof. Yaping Shao, we also think that some measurements are necessary for validation of our simulation results. Unfortunately, the measurement of snowdrift sublimation in the saltation layer is very difficult to conduct at the present stage.

**could you check equation (14) (or may be I did a mistake but molecular weight of water have to be taken into account in the first term of the denominator) ? Some details concerning some threshold or constant as the one for take-off have to be given as they are crucial.**

**Reply**: Thanks. Indeed, the first term of the denominator in the sublimation rate given by Thorpe and Mason is related to the molecular weight of water M (kg mol$^{-1}$), as well the universal gas constant R (J mol$^{-1}$ K$^{-1}$). But in our study, the gas constant for water vapor $R_V$ (=R/M) is defined with the value of 461.5 (J kg$^{-1}$ K$^{-1}$). According to the experiments and numerical simulation of Nemoto and Nishimura (2001, 2004), we set the threshold friction velocity of snow to be 0.21 m s$^{-1}$.

**However the aim of this work is interesting and also the way to treat it.**

**Reply**:Thanks for the positive evaluation of our work.

*Referee #3*

**In this study, the authors present an interesting research to evaluate the effect of drafting snow sublimation in the saltation layer, which is rarely studied, but important for the hydrological balance of snow cover. A snowdrift model with considering the coupling effects of snow sublimation, temperature, humidity and moisture transport is established to address their ideas and research. They describe their models and methods concisely and clearly and analyze the results reasonably.**

**Reply**: We thank the reviewer for the positive evaluation of our work.

**They present a well-written article, but the Results section seems a little short and has no comparison with the measurements or published data.**

**Reply**: Just like the reviewers and Prof. Yaping Shao, we also think that some measurements are necessary for validation of our simulation results. However, the measurement of snowdrift sublimation in the saltation layer is very difficult to conduct at the present stage. We have tried our best to make our results comparison with the published studies. Unfortunately, suitable data for comparison were not found. We demonstrated the importance of snowdrift sublimation in the saltation layer from numerical simulation and honestly suggest relevant measurement should be carried out in future work.

**The authors need to carefully check their manuscript again to do a better job of defining the parameters they use for the controlling equations, for example, the definition of the sublimation rate S.**

**Reply**: Thanks. The sublimation rate $S$ at each height is calculated by summed over the mass loss of all particles at each height above the surface. Theoretically, it is negative, but here taken as positive for illustration purposes. We have pointed that in the revised manuscript.

**The prognostic equations of potential temperature and specific humidity (Eqs. 11 and 12) seem to be different from the reference, which may lead to misunderstandings (just as the comments of reviewer 1). In order to express the calculation of temperature and humidity more clearly, it would be better to derive the prognostic equations from the basic convection diffusion equation.**

**Reply**: Thanks for the insightful comment. Following the reviewer's suggestion, we have derived

the prognostic equations of potential temperature and specific humidity (Eqs. (13) and (14)) from the basic convection diffusion equation (Eqs. (11) and (12)) in the revised manuscript. In addition, we stated that three cases considered in our study to explore the interaction between snow sublimation and moisture transport, i.e. two typical cases, neglecting the effects of moisture transport and considering moisture transport due to both moisture diffusion and advection, and a common situation, considering only moisture diffusion.

**Why each calculation takes 60 s and doesn't show the sublimation rate when it tends to be stabilized?**

**Reply**: Thanks for the insightful comment. By considering of the required time of drifting snow development and the capability of computer, the simulated time was set as 60s, which is significantly surpass drifting snow development time (about 2-3 s) and could be actualized easily on PC. From Fig. 3, we can see that DSS has reached steady state with moisture diffusion and advection considered within 60 s, but it is not true for only moisture diffusion considered. That is indeed that, the sublimation rate is hard to be stabilized in this 60s in this case, but our results could expose the issues that we care about. Theoretically, the snow sublimation may vanish finally with the development of time. As we state in our manuscript, DSS has an inherent self-limiting nature due to the feedback associated with the heat and moisture budgets. On one hand, snow sublimation absorbs heat, which decreases the temperature of the ambient air and the saturation vapor pressure; on the other hand, it will induce the increment in the moisture content of the ambient air. Both of the above two points can increase the relative humidity of ambient air. It may lead to a saturated layer near the surface finally, and thus sublimation will vanish. Here, we show the temporal evolution of drifting snow sublimation within 60 s to compare our results with the previous study.

**I agree that it could be discussed if the variation of temperature could induce some effects on velocity field. The evolutions of snow particles concentration should be discussed.**

**Reply**: Thanks. Indeed, the variation of temperature could induce some effects on velocity field. However, we found that this effect can be ignored by testing (Fig. 1.1). In our study, the variation of temperature due to snow sublimation is below 2 K, which is relatively low. From Figure 1.1, we can see that the effect of variation of temperature on velocity field is very small. Thus, we didn't take this effect into consideration. In addition, following the reviewer's suggestion, we added a figure to show snow particles concentration profile evolution in the revised manuscript.

**On the whole, this work is a fundamental research and obviously focusing on science**

**common topics, which is interesting and well present.**

**Reply**: Thanks for the positive evaluation of our work.

**List of changes**

**Page 1**

Line 4: "N. Huang and X. Dai" is changed to "N. Huang[1, 2],X. Dai[1] and J. Zhang[1, 2]".

We add J. Zhang as the third author for his contribution on program test and improvement of the revised manuscript.

Line 7: "[2]School of Civil Engineering and Mechanics, Lanzhou University, Lanzhou 730000, China" is added before "Corresponding to".

**Page 4**

Line 14: "we followed previous researches to assume relative humidity adjacent to snow surface is saturated and ignored surface sublimation. But the particle sublimation in saltation layer is considered by taking into account of moisture transport in different typical cases, including 1) neglecting the effects of moisture transport; 2) considering moisture transport due to both moisture diffusion and advection, and 3) considering only moisture diffusion. Here," is added after "In this study".

**Page 5**

Line 14: "is" is changed to "satisfies".

**Page 6**

Line 15: "(the ratio of vertical ejection velocity and vertical impact velocity)" is added after" $e_v$ ".

Line 18: "(the ratio of horizontal ejection velocity and horizontal impact velocity)" is added after " $e_h$ ".

Line 23- Line 12 (Page 7): "following prognostic equations…with $l$ being the length of the domain." is changed to "conservation equations (only consider two-dimension)

$$\frac{\partial \theta}{\partial t} + u\frac{\partial \theta}{\partial x} + v\frac{\partial \theta}{\partial y} = \frac{\partial}{\partial x}\left(K_{\theta'}\frac{\partial \theta}{\partial x}\right) + \frac{\partial}{\partial y}\left(K_{\theta}\frac{\partial \theta}{\partial y}\right) + R_1 \tag{11}$$

$$\frac{\partial q}{\partial t} + u\frac{\partial q}{\partial x} + v\frac{\partial q}{\partial y} = \frac{\partial}{\partial x}\left(K_{q'}\frac{\partial q}{\partial x}\right) + \frac{\partial}{\partial y}\left(K_{q}\frac{\partial q}{\partial y}\right) + R_2 \tag{12}$$

where $u$ is the mean horizontal wind velocity which could be calculated by Eq. (5) and $v$ the vertical wind velocity is assumed to be zero here; $K_{\theta'}$, $K_{\theta}$, $K_{q'}$ and $K_q$ are the heat and moisture diffusivities due to molecular motion and eddy diffusivity, respectively ; $R_1$ and $R_2$ are the source terms due to snow sublimation. In this study, the wind speed is parallel to the horizontal direction, moreover, we hypothesize that the temperature and specific humidity is linearly distributed along this direction. Thus, potential temperature and specific humidity will satisfy the following prognostic equations

$$\frac{\partial \theta}{\partial t} = \frac{\partial}{\partial y}\left( K_\theta \frac{\partial \theta}{\partial y} \right) - u\frac{\partial \theta}{\partial x} - \frac{L_s S}{\rho_f C} \tag{13}$$

$$\frac{\partial q}{\partial t} = \frac{\partial}{\partial y}\left( K_q \frac{\partial q}{\partial y} \right) - u\frac{\partial q}{\partial x} + \frac{S}{\rho_f} \tag{14}$$

where $K_\theta = \kappa u_* y + K_T$ and $K_q = \kappa u_* y + K_V$ (the sum of eddy diffusivity and molecular diffusivity, respectively); $S$ is the sublimation rate summed over all particles at each height above the surface, here taken as positive for illustration purposes; $L_s$ is the latent heat of sublimation ($2.835 \times 10^6$ J kg$^{-1}$); $C$ is the specific heat of air; $\frac{\partial \theta}{\partial x}$ and $\frac{\partial q}{\partial x}$ represent the horizontal gradient in temperature and specific humidity. At the edge of snow surface, we considered the effect of advection and hypothesized that the specific humidity in the study domain is linearly distributed along the horizontal direction from entrance with $q_{in}$ to outlet with $q_{out}$. Thus, the horizontal advection of moisture can be simplified to $u(q_{out} - q_{in})/l$, with $l$ being the length of the domain. Except for snow surface edge, the above setup may be (or partly) suitable for some heterogeneous snow surfaces, such as patchy mosaic of snow cover. And these reasons encourage us to discuss the effect of advection. For the case of infinite and homogenous snow surface, we set $q_{in} = q_{out}$ to avoid advection and considered moisture transfer via molecular motion and eddy diffusivity. Besides, we set $q_{in} = q_{out}$ and $K_q = K_\theta$ to ignore effect of advection and eddy diffusivity, as a reference case. Correspondingly, similar process was actualized for $\theta$. The variation of temperature will induce some effects on velocity field, which, however, can be ignored by testing. In our study, the variation of temperature due to snow sublimation is relatively low and its effect on velocity field is very small. Thus, we didn't take this effect into consideration."

**Page 7**

Line 15: "(13)" is changed to "(15)".

Line 18: "radius $r$ " is changed to "diameter $D$ ".

Line 19: Eq. (14) is changed to " $\dfrac{dm}{dt} = \dfrac{\pi D\delta}{\dfrac{L_s}{KTNu}(\dfrac{L_s}{R_v T}-1)+\dfrac{R_v T}{D_v She_s}}$  (16)".

**Page 8**

Line 4: "(15)" is changed to "(17)".

Line 8: "(16)" is changed to "(18)".

Line 11: "(17)" is changed to "(19)".

Line 15: "(18)" is changed to "(20)".

Line 18: "(19)" is changed to "(21)".

**Page 9**

Line 3- Line 4: "(16) - (19)" is changed to "(18) - (21)".

Line 6- Line 7: The sentence "The threshold friction velocity of snow is 0.21 m s$^{-1}$ and the snow bed roughness is $3.0 \times 10^{-5}$ m (Nemoto and Nishimura, 2001)." is changed to "According to the investigation of Nemoto and Nishimura (2001) in a cold wind tunnel, the threshold friction velocity of snow is set to be 0.21 m s$^{-1}$ and the snow bed roughness $3.0 \times 10^{-5}$ m".

Line 24: "(13) - (15)" is changed to "(15) - (17)".

**Page 10**

Line 1: "3.1 Relative Humidity and Temperature" is changed to "3.2 Relative Humidity and Temperature", "Wind-blown snow has a self-regulating feedback mechanism between the saltating particles and the wind field, i.e. snow particles are entrained and transported by the wind, while the drag force associated with particle acceleration reduces the wind velocity in the saltation layer, thus limiting the entrainment of further particles. Figure 1 illustrates the evolution of saltating snow particles in air and also the profile of snow particle number density at steady state. The results show that the transport rate of particles in air increases rapidly and reaches a steady state after 2-3 seconds. In steady condition, the number of snow particles decreases with height and follows a negative exponential law. Except for the particle in air, the ambient relative humidity and temperature are also important factors concern to DSS." is added before "3.2 Relative Humidity and Temperature".

Line 3: "Fig. 1a" is changed to "Fig. 2a".

Line 4: "Fig. 1b" is changed to "Fig. 2b".

Line 6: "continually" is deleted.

Line 8: "Fig. 1" is changed to "Fig. 2".

Line 10: "when moisture diffusion is included" is added after "increase", "Fig. 1a" is changed to "Fig. 2a".

Line 18: "Fig. 2" is changed to "Fig. 3".

Line 19: "When the advection of moisture and heat are considered as well, the temperature and relative humidity will reach a steady state finally. In this case, the transport of moisture and heat balances the change of temperature and relative humidity due to DSS." is added at the end of the paragraph.

Line 20: "3.2" is changed to "3.3".

Line 21: "Fig. 2" is changed to "Fig. 3", "From Fig. 2, we can see that DSS has reached steady state with moisture diffusion and advection considered within 60 s, but it is not true for only moisture diffusion considered. By considering of the required time of drifting snow development and the capability of computer, the simulated time was set as 60s, which is significantly surpass drifting snow development time (about 2-3 s) and could be actualized easily on PC. Furthermore, the results are enough to expose the issues that we care about." is added at the beginning of the paragraph.

Line 27: "and will reach steady state" is added after "reduced".

Line 30: "$0.61 \times 10^{-5}$" is changed to "$0.88 \times 10^{-5}$".

Line 32: "$0.96 \times 10^{-5}$" is changed to "$1.6 \times 10^{-5}$", "0.83" is changed to "1.38".

**Page 11**

Line 2: "Fig. 2" is changed to "Fig. 3".

Line 7: "saturation" is changed to "unsaturation".

Line 10: "Fig. 3" is changed to "Fig. 4".

Line 12: "(Figure 1)" is added after "heights".

Line 14: "number" is added before "density".

Line 19: "Fig. 3" is changed to "Fig. 4" and "$10^{-4}$" is changed to "$10^{-4}$ - $10^{-3}$".

**Page 16**

A new figure is added.

[revised manuscript text omitted]